# Strategic Feature Selection

## Abstract

Algorithmic prediction rules are increasingly used to allocate resources, such as targeting households for social welfare programs, determining payments to Medicare Advantage insurers, and assigning eligibility for social benefits, all of which create incentives for strategic manipulation of input features. Policymakers often respond by excluding manipulable features from the prediction model, yet it is not well understood when this reduces the prediction risk. In this paper, we analyze feature selection under strategic behavior in a linear regression setting, motivated by risk adjustment models in U.S. health policy. Our model characterizes how organizations strategically manipulate reported features in response to decision rules, and how a regulator can counteract such strategic behavior through feature selection. We establish sufficient conditions on the cost structure of feature manipulation that identify when excluding manipulable features reduces prediction risk, and conversely, when retaining the full feature set yields more accurate predictions. These results offer a first step toward principled feature selection methods that explicitly account for unreliable and strategically manipulated data inputs.

## 1 Introduction

Algorithmic predictions are increasingly used to inform decision-making about allocation of resources. Decision-makers rely on individuals' *features* to determine eligibility and set allocation amounts, with the aim of implementing normative priorities. For example, eligibility for social welfare programs is determined using poverty-targeting scores [2, 23], and government payments to health providers and insurers is based on patient risk scores [7]. Such algorithmic decision-making systems incentivize organizations that serve individuals to respond strategically and "game" the prediction rule.

We consider the U.S. Medicare Advantage (MA) program as a running example, where the government determines payments to private insurers using a public risk-adjustment model that is trained to predict patient costs given health data from the previous year [5, 19]. The goal of risk adjustment is to ensure that insurers receive higher payments for higher-risk enrollees who are expected to need more services. This payment rule inadvertently introduces incentives for private insurers to overreport diagnosis codes, thereby inflating risk-adjusted payments, a practice known as "upcoding." In 2024, higher MA risk scores were estimated to translate into $50 billion in overpayments, as a result of upcoding [15].

To counteract the effect of upcoding, Centers for Medicare & Medicaid Services (CMS) excludes diagnoses that are at risk of inappropriate coding by health plans and providers [3, 4]. In 2024, CMS removed the conditions corresponding to Protein-Calorie Malnutrition and Angina Pectoris from the payment model to limit the sensitivity of the model to higher coding intensity in MA and maintain the ability to accurately predict costs [4]. Despite the use of feature selection as a policy lever to combat manipulation, it remains difficult to reason about which features a decision-maker should exclude in response to strategic behavior, since dropping features comes at the cost of predictive accuracy.

Submitted to 39th Conference on Neural Information Processing Systems (NeurIPS 2025). Do not distribute.

To address this gap, we develop a formal framework to reason about feature selection under strategic behavior. We build on existing frameworks of strategic learning [9], but with a focus on policy levers commonly used in practice that are perhaps more coarse and simple, but as a result more widely applied. In addition, while general strategic learning requires detailed information about costs to manipulation, we focus on realistic limited information settings.

**Contributions.** We present a theoretical model of a decision-maker's choice to drop or retain features in a prediction model when such features can be strategically manipulated. We focus on a regression setting, which aligns with the risk-adjustment models used by CMS. We give sufficient conditions for the decision-maker to be better off dropping or retaining features, which we also pair with simulations and examples. Finally, we discuss future directions towards practical policy recommendations.

## 1.1 Related work

A growing line of work in strategic classification is aimed at learning optimal prediction rules when decision subjects can manipulate their features at a cost [9, 6, 22]. Different from prior literature that focused on pure *gaming*, the goal in [18, 12, 22] is to additionally incentivize genuine improvement in individual outcomes. [8, 22, 11] study settings in which individuals have hidden features that causally affect the outcome. In contrast, we study a setting where the decision-maker explicitly excludes manipulable features from the prediction rule. Closest to our work is that of Holmstrom and Milgrom [11]. An important difference is the decision-maker's prediction risk objective in our work.

The health policy literature has put forth several concrete proposals for feature selection in Medicare Advantage risk adjustment. These proposals suggest, for example, including patient survey data [1, 14] or excluding diagnoses added to a patient health record via chart review [17]. Our work is distinct in that it provides a principled framework to navigate a set of feature selection decisions.

## 2 Model and Problem Formulation

We study the strategic interaction between organizations that receive predictions based on the features of the individuals they serve, and a decision-maker who specifies the prediction rule. The decision-maker publishes a prediction rule $f_\theta : \mathcal{X} \to \mathcal{Y}$ parameterized by $\theta \in \Theta \subseteq \mathbb{R}^d$, mapping an individual's features $x \in \mathcal{X} \subseteq \mathbb{R}^d$ to the predicted outcome $\hat{y} \in \mathcal{Y} \subseteq \mathbb{R}$. We consider linear prediction rules $f_\theta(x) = \theta^\top x = \sum_{i=1}^d \theta_i x_i$ such that $\theta_i \geq 0$ for all $i$. The linear specification is a commonly studied setting in strategic learning [10, 12, 22]. Moreover, this is not merely a modeling assumption: many models used by CMS in practice—including the risk adjustment model used to allocate payments to Medicare Advantage plans—are based on least squares regression [20, 21].

The organization observes features $x$ corresponding to an individual and takes action $a \in \mathcal{A} \subseteq \mathbb{R}^d$ to manipulate the features from $x$ to $x + a$ in order to maximize the predicted outcome $\hat{y}$. We model the population of individuals as a distribution $\mathcal{P}_0$ over the space $\mathcal{Z} = \mathcal{X} \times \mathcal{Y}$ with mean $\mu = \mathbb{E}_{x \sim \mathcal{P}_0}[x]$ and covariance $\Sigma$. We consider the distribution of individuals to be the same across organizations and study a single type of organization in this work.

We assume that the decision-maker has access to *unmanipulated* data points from the space $\mathcal{Z} = \mathcal{X} \times \mathcal{Y}$. Such data are obtained either from a pre-deployment period (prior to the use of $f_\theta$) or from an alternative policy under which organizations are not incentivized to manipulate their features. In Medicare, this corresponds to traditional Medicare (fee-for-service), where payments are not based on enrollees' risk scores and thus there is no incentive to overreport diagnoses [15]. We consider the true outcome model is a linear function $y = \theta^{*\top} x$ and the decision-maker estimates the true parameters $\theta^*$ using unmanipulated samples from $\mathcal{Z}$.

**Organization's best-response model.** The organization incurs a cost $C(a) > 0$ for modifications resulting from action $a$. We model the cost function as $C(a) = \frac{1}{2} a^\top H a$, where $H \in \mathbb{R}^{d \times d}$ is the cost matrix and $H \succ 0$. For private insurers, this cost arises from payments to chart review contractors for mining additional diagnosis codes and from conducting in-home health assessments aimed at identifying undocumented conditions [7]. We assume that organizations behave rationally and *best-respond* to $f_\theta$ by choosing action

$$a^*(x; \theta) = \arg\max_{a \in \mathcal{A}} \ \theta^\top (x + a) - \frac{1}{2} a^\top H a \qquad (1)$$

86  We can compute the action the organization will take by maximizing (1) over $a \in \mathcal{A}$. Although
87  real-world settings sometimes restrict $\mathcal{A}$ (e.g., binary or bounded actions), in this work we consider
88  the unconstrained case $\mathcal{A} = \mathbb{R}^d$. Note that $\nabla_a(\theta^\top(x + a) - \frac{1}{2}a^\top H a) = \theta - Ha$. If $\mathcal{A} = \mathbb{R}^d$ and
89  $H \succ 0$, $a^*(x; \theta) = H^{-1}\theta$. From here on, we omit the dependence on $x$ and write $a^*(\theta) = H^{-1}\theta$.

90  **Decision-maker's objective.** The decision-maker's goal is to predict the true outcome as accurately
91  as possible. For example, the government seeks to avoid over- or under-estimating enrollees' risk,
92  thereby minimizing corresponding over- and under-payments to insurers. We define prediction
93  risk under strategic response as mean squared error $\text{MSE}(\theta)$, and the decision-maker chooses $\theta$ to
94  minimize $\text{MSE}(\theta)$:

$$\text{MSE}(\theta) = \mathbb{E}_{x \sim \mathcal{P}_0}\left[(\theta^\top(x + a) - \theta^{*\top}x)^2\right] \tag{2}$$

95  When the decision-maker has full information of $H, \mathcal{P}_0$, and $\theta^*$, the minimum MSE is defined
96  as the "strategic optimum" if we think of our model as a game between the organization and the
97  decision-maker.

98  **Definition 2.1** (Strategic optimum)**.** The *strategic optimum* in the full-information game is defined as

$$\text{OPT}_{\theta^*}(\mathcal{P}_0, H) = \min_{\theta \in \Theta} \mathbb{E}_{x \sim \mathcal{P}_0}\left[(\theta^\top(x + a^*(\theta)) - \theta^{*\top}x)^2\right]$$

99  In the real world, however, decision-makers do not have full information about the cost structure.
100 They often rely on simple—but not formally justified—heuristics, as we discuss in the next section.

## 2.1  $\text{MSE}_{\text{full}}$ and $\text{MSE}_{\text{drop}}$

102 In practice, decision-makers often drop features they expect to be highly manipulable, as noted earlier
103 with diagnoses in the CMS risk-adjustment model [4]. Yet, the conditions under which doing so
104 reduces the decision-maker's prediction risk are not well understood. We characterize regimes in
105 which feature dropping lowers risk and regimes in which the full model is preferable. Specifically,
106 we identify conditions under which it is optimal for the decision-maker to retain all features.

107 For the remainder of this work, we study the two feature case $X = (X_1, X_2)$ and focus on the model
108 that drops $X_2$. The analysis for dropping $X_1$ is analogous. For simplicity, we assume a diagonal cost
109 matrix i.e., $H^{-1} = \text{diag}(h_{11}, h_{22})$ and $h_{12} = 0$. In this case, $y = \theta_1^* x_1 + \theta_2^* x_2$ and we denote mean
110 $\mu = (\mu_1, \mu_2)$ and second-moment matrix $M = \mathbb{E}_{x \sim \mathcal{P}_0}[xx^\top]$.

111 The decision-maker learns $\hat{\theta}$ using samples $(x, y)$ drawn from $\mathcal{P}_0$ by minimizing the risk

$$\hat{\theta} = \arg\min_{\theta \in \Theta} \mathbb{E}_{(x,y) \sim \mathcal{P}_0}\left[(\theta^\top x - y)^2\right]. \tag{3}$$

112 From the definition of our linear model, $\hat{\theta}_{\text{full}} = \theta^*$ in the full feature model. We start by computing
113 the MSE of the full feature model ($\text{MSE}_{\text{full}}$) as

$$\text{MSE}_{\text{full}} = \mathbb{E}_{x \sim \mathcal{P}_0}\left[(\hat{\theta}_{\text{full}}^\top(x + H^{-1}\hat{\theta}_{\text{full}}) - \theta^{*\top}x)^2\right] = (\theta^{*\top}H^{-1}\theta^*)^2 = \left(\theta_1^{*2}h_{11} + \theta_2^{*2}h_{22}\right)^2. \tag{4}$$

114 This follows from substituting the best-response of the organization. When the decision-maker
115 chooses to drop $X_2$, they learn a single parameter $\hat{\theta}_{\text{drop}} = (\beta_1, 0)$. We derive $\text{MSE}_{\text{drop}}$ in Lemma A.1
116 and directly state here:

$$\text{MSE}_{\text{drop}} = (h_{11}\beta_1^2 - \theta_2^*(\mu_2 - r\mu_1))^2 + \theta_2^{*2}(\Sigma_{22} + r^2\Sigma_{11} - 2r\Sigma_{12}), \tag{5}$$

117 where $r := M_{12}/M_{11}$. From here on, for brevity, we will use $\Delta := \mu_2 - r\mu_1$ and $V := \Sigma_{22} +$
118 $r^2\Sigma_{11} - 2r\Sigma_{12}$. Note that both $\text{MSE}_{\text{full}}, \text{MSE}_{\text{drop}} \geq \text{OPT}_{\theta^*}(\mathcal{P}_0, H)$.

## 3  Results

120 In this section, we give sufficient conditions for when dropping features is better than keeping all
121 features, and vice versa. All proofs are deferred to Appendix A.

122 First, we show that there exists a high-cost regime (i.e., $h_{11}, h_{22}$ are small) where retaining both
123 features strictly dominates any one-feature model.

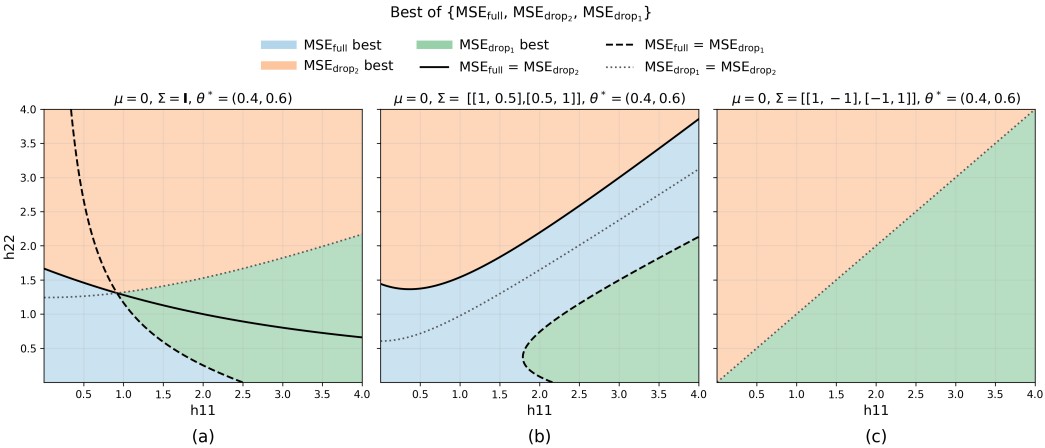

Figure 1: **Best model across manipulation costs**. For each $(h_{11}, h_{22})$ (entries of $H^{-1}$), the shading shows which model yields the lowest MSE. Boundary curves mark equal-risk frontiers.

**Proposition 3.1** (Features can be retained at high cost). *If the covariance matrix $\Sigma$ is positive definite, or $\Delta \neq 0$, then there exists $\varepsilon > 0$ such that for any $h_{11}, h_{22} < \varepsilon$, $\mathrm{MSE}_{\mathrm{full}} < \mathrm{MSE}_{\mathrm{drop}}$.*

The main message of this result is that it is beneficial for the decision-maker to safely retain all features whenever manipulation is kept sufficiently costly. Specific to our example, a central policy lever CMS can use to disincentivize upcoding is to increase the cost of manipulation—via higher legal penalties and more comprehensive audits. We examine this regime in Figure 1 for (a) independent and (b) correlated features, where the region with dominance of $\mathrm{MSE}_{\mathrm{full}}$ is shaded in blue.

When $\Delta = 0$ and $V = 0$ ($\Sigma$ has rank 1), it is possible that there exists no cost setting, no matter how high, in which the full model dominates the one-feature model. We give a specific example of feature distributions for which this happens in Figure 1 (c). We note that when manipulation is cheap, $\Sigma$ determines the winner (Figure 1 (a, b)). This establishes that feature correlation should guide feature selection decisions. It also bears out the intuition that health policymakers have long stood by: as you include more features, you expand the "gameable" surface area of the model [13].

Next, we show that the difference $\mathrm{MSE}_{\mathrm{drop}} - \mathrm{MSE}_{\mathrm{full}}$ is monotone when $h_{11}, h_{22} \geq 0$.

**Proposition 3.2** (Unique $h_{22}$-threshold for drop vs full). *For any given $h_{11} > 0$, there exists a unique threshold $h_{22}^* \in \mathbb{R}$ such that*

$$\mathrm{sgn}(\mathrm{MSE}_{\mathrm{drop}} - \mathrm{MSE}_{\mathrm{full}}) = \begin{cases} +1, & \text{for } 0 < h_{22} < h_{22}^*, \\ 0, & \text{for } h_{22} = h_{22}^*, \\ -1, & \text{for } h_{22} > h_{22}^*. \end{cases}$$

In this case, given an estimate for $h_{11}$, the decision-maker can compute $h_{22}^*$ and make a choice from only one-sided information on $h_{22}$. Moreover, this points to a more efficient feature-level auditing strategy, rather than auditing at the patient level. We provide additional simulations in Appendix B with different values of $(\mu_1, \mu_2)$, $\Sigma$ and $\theta^*$ and show that our results are consistent.

## 4 Discussion

We present a model to formally reason about feature selection when such features can be strategically manipulated. In particular, we provide sufficient conditions under which the decision-maker should drop features rather than retain the full set, and conversely when the full model is optimal. From a policy standpoint, an important next step is to identify by how much the manipulation costs should be raised to provide concrete recommendations. An interesting direction for future work is to study a setting where organizations differ in the population of individuals they serve. Further, organizations could face different costs of manipulation. For example, Medicare Payment Advisory Commission [16] has found substantial heterogeneity in coding intensity across MA organizations. Amid a growing body of work on strategic classification, we hope our work invites further investigation of feature selection under strategic behavior.

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

 # A  Proofs

**Lemma A.1** (MSE$_{\text{drop}}$)**.** *Let $X = (X_1, X_2)$ with mean $\mu = (\mu_1, \mu_2)$ and covariance $\Sigma$, $Y = \theta_1^* X_1 +$*
*$\theta_2^* X_2$. When the decision-maker chooses to drop $X_2$, they learn a single parameter $\hat{\theta}_{\text{drop}} = (\beta_1, 0)$.*
*In this case, the mean–squared error equals*

$$\text{MSE}_{\text{drop}} = (h_{11}\beta_1^2 - \theta_2^*(\mu_2 - r\mu_1))^2 + \theta_2^{*2}(\Sigma_{22} + r^2\Sigma_{11} - 2r\Sigma_{12})$$

*Proof.* We can compute $\beta_1$ as

$$\beta_1 = \arg\min_{\beta} \mathbb{E}_{(X,Y) \sim \mathcal{P}_0} \left[ (\beta X_1 - Y)^2 \right]$$

$$= \frac{\mathbb{E}[X_1 Y]}{\mathbb{E}[X_1^2]}$$

$$= \theta_1^* + \theta_2^* \frac{\mathbb{E}[X_1 X_2]}{\mathbb{E}[X_1^2]}$$

$$= \theta_1^* + \theta_2^* r,$$

where $r := M_{12}/M_{11}$, $M := \mathbb{E}[XX^\top] = \Sigma + \mu\mu^\top$.

The organization's best response to $\hat{\theta}_{\text{drop}}$ is $H^{-1}\hat{\theta}_{\text{drop}} = (h_{11}\beta_1, 0)$.

Let $\varepsilon_{\text{drop}} := (\hat{\theta}_{\text{drop}}^\top (X + H^{-1}\hat{\theta}_{\text{drop}}) - \theta^{*\top} X)$. Then,

$$\varepsilon_{\text{drop}} = h_{11}\beta_1^2 + \beta_1 X_1 - \theta_1^* X_1 - \theta_2^* X_2$$

$$= h_{11}\beta_1^2 - \theta_2^*(X_2 - rX_1)$$

Thus,

$$\mathbb{E}_{X \sim \mathcal{P}_0}[\varepsilon_{\text{drop}}] = h_{11}\beta_1^2 - \theta_2^*(\mu_2 - r\mu_1)$$

$$\text{Var}(\varepsilon_{\text{drop}}) = \theta_2^* \text{Var}(X_2 - rX_1)$$

$$= \theta_2^* (\Sigma_{22} + r^2\Sigma_{11} - 2r\Sigma_{12})$$

Since

$$\text{MSE}_{\text{drop}} = \mathbb{E}_{X \sim \mathcal{P}_0}[\varepsilon_{\text{drop}}^2] = (\mathbb{E}[\varepsilon_{\text{drop}}])^2 + \text{Var}(\varepsilon_{\text{drop}}),$$

the result follows. $\qquad\qquad\square$

*Proof of Proposition 3.1.* $\text{MSE}_{\text{full}} < \text{MSE}_{\text{drop}}$ if

$$\left(\theta_1^{*2}h_{11} + \theta_2^{*2}h_{22}\right)^2 < (h_{11}\beta_1^2 - \theta_2^*\Delta)^2 + \theta_2^{*2}V.$$

This is equivalent to the inequality

$$(\theta_1^{*4} - \beta_1^4)h_{11}^2 + \theta_2^{*4}h_{22}^2 + 2\theta_1^{*2}\theta_2^{*2}h_{11}h_{22} + 2\beta_1^2\theta_2^*\Delta h_{11} < (\theta_2^*\Delta)^2 + \theta_2^{*2}V.$$

The right hand side of the above inequality is strictly positive, since either the covariance matrix $\Sigma$ is
PD, or $\Delta \neq 0$. The left hand side tends to zero continuously as $h_{11}, h_{22}$ tend to zero.

Specifically, choose $\delta > 0$ such that $(\theta_2^*\Delta)^2 + \theta_2^{*2}V > \delta$. Choose $\varepsilon$ such that $(\theta_1^{*4} - \beta_1^4 + \theta_2^{*4} +$
$2\theta_1^{*2}\theta_2^{*2})\varepsilon^2 + 2\beta_1^2\theta_2^*\Delta\varepsilon < \delta$. $\qquad\qquad\square$

*Proof of Proposition 3.2.* Fix $h_{11} > 0$. From (4) and (5), we know

$$g(h_{22}) = \text{MSE}_{\text{drop}} - \text{MSE}_{\text{full}} = \left(h_{11}\beta_1^2 - \theta_2^*\Delta\right)^2 + \theta_2^{*2}V - (\theta_1^{*2}h_{11} + \theta_2^{*2}h_{22})^2$$

Let $D(h_{11}) := \left(h_{11}\beta_1^2 - \theta_2^*\Delta\right)^2 + \theta_2^{*2}V \,(\geq 0)$. Then, $g(h_{22}) = D(h_{11}) - (\theta_1^{*2}h_{11} + \theta_2^{*2}h_{22})^2$ is
strictly decreasing on $[0, \infty)$. For $h_{22} \geq 0$,

$$g'(h_{22}) = -2\theta_2^{*2}(\theta_1^{*2}h_{11} + \theta_2^{*2}h_{22}) < 0$$

As $h_{22} \to \infty$, $g(h_{22}) \to -\infty$. At $h_{22} = 0$, $g(0) = D(h_{11}) - (\theta_1^{*2} h_{11})^2$. If $g(0) \geq 0$, there exists a unique $h_{22}^*$ with $g(h_{22}^*) = 0$. We solve

$$D(h_{11}) = (\theta_1^{*2} h_{11} + \theta_2^{*2} h_{22}^*)^2, \quad \theta_1^{*2} h_{11} + \theta_2^{*2} h_{22}^* \geq 0$$

to get the threshold

$$h_{22}^* = \frac{-\theta_1^2 h_{11} + \sqrt{D(h_{11})}}{\theta_2^2}.$$

Note that the negative square-root branch is inadmissible. Since $g(h_{22})$ is strictly decreasing, we get the desired result.

If $g(0) < 0$ (i.e., $\sqrt{D(h_{11})} < \theta_1^{*2} h_{11}$), $g(h_{22}) < 0$ for any $h_{22} > 0$. Thus, the threshold $h_{22}^* \leq 0$. In this case, the interval $(0, h_{22}^*)$ is empty and the result still holds.

$\square$

## B   Additional simulations

We run additional simulations varying $\mu, \Sigma$, and $\theta^*$ and present the results in Figure 2.

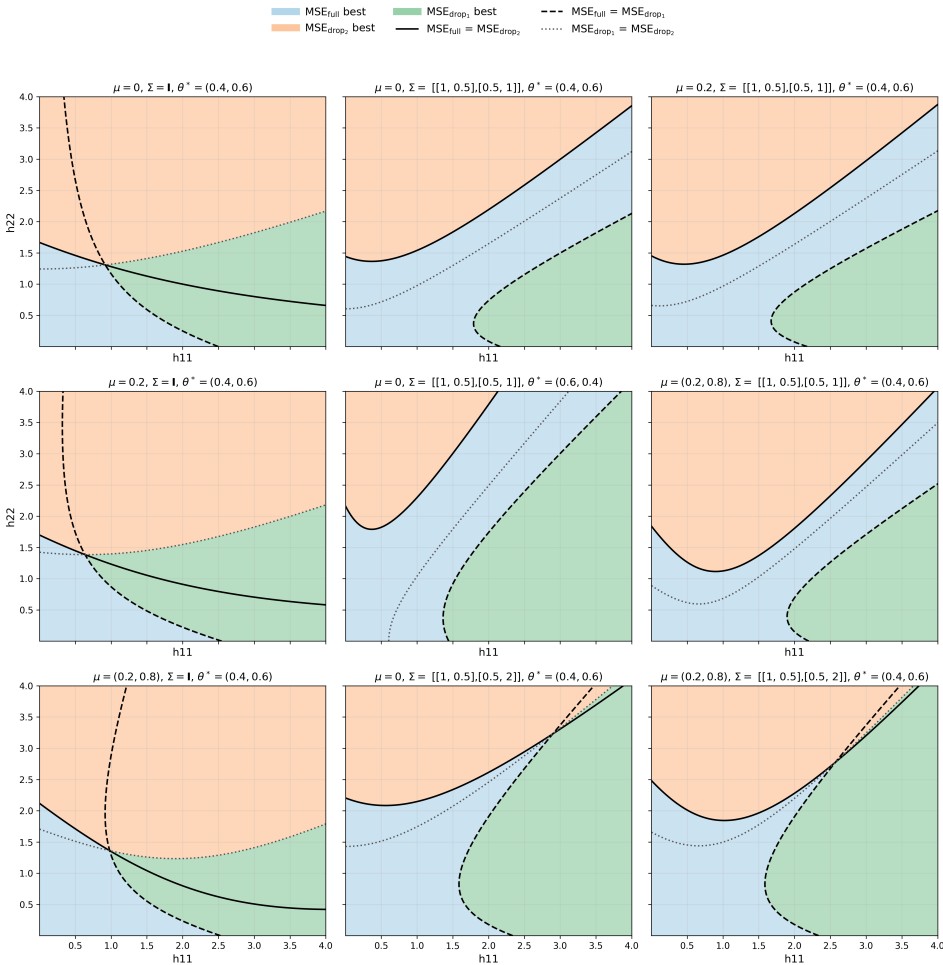

Figure 2: **Best model across manipulation costs**. For each $(h_{11}, h_{22})$ (entries of $H^{-1}$), the shading shows which model yields the lowest MSE. Boundary curves mark equal-risk frontiers. Top labels for each plot indicate $\mu, \Sigma$, and $\theta^*$.

