# OpenReview forum: "Strategic Feature Selection"
_NeurIPS.cc/2025/Workshop/Reliable_ML — NeurIPS 2025 - Reliable ML Workshop_

### Official Review · Reviewer_P397 · 2025-09-13

**Rating:** 3
**Confidence:** 3

**Review:**

**Summary:**

The authors consider a setting where a learner is given feature vectors $x \in \mathbb{R}^d$ that have been manipulated by a strategic adversary, and is tasked predicting the labels $y \in \mathbb{R}^d$. In more detail, the learner first selects an algorithm for choosing a prediction rule and reveals it to the adversary. The adversary then samples feature vectors from some underlying distribution and can incur a cost to manipulate them, before revealing them to the learner. Their aim is to maximize a utility function, which depends both on the learner’s predictions and on the cost incurred to manipulate the sampled datapoints.

In this paper, the authors examine the case where the learner’s prediction rule is linear regression $y = \theta^\top x$, and the learner has the following choice regarding how to choose $\theta$: they must pick a subset $T \subset [d]$ of coordinates, set $\theta_T = 0$, and then pick the rest of the coordinates to minimize the MSE $\mathbb{E} [(\theta_{S}^\top x_{S} - y)^2]$, where $S = [d] \setminus T$. That is, the learner can choose to ignore any number of features in their regression model, potentially disincentivizing the adversary from manipulating the data.

**Strengths:**

The setting of the paper is well-motivated, as the authors present real-life examples (particularly related to health insurance) where the party providing the data has strong incentive to falsely report them, and the party receiving them drops features that are considered highly manipulable. The paper is easy to read and the presentation is clean.

**Weaknesses/Limitations:**

Overall, the paper is lacking in technical novelty and impact. Too many simplifying assumptions are made on the model, specifically:

* The learner is given perfect knowledge of the true distribution of $(x, y)$, thus the adversary cannot give poisoned samples to the learner to affect the prediction rule that they choose.
* The adversary’s gain for a given feature vector $x$ is simply the value of $y$ that the learner predicts for that $x$. The cost that the adversary incurs for manipulating a feature vector $x$ into a vector $x + a$ is just a quadratic $\frac{1}{2} a^\top H a$ for some PSD matrix $H$. Therefore, the total utility of the adversary for a given $x$ is equal to $\theta^\top (x + a) - \frac{1}{2} a^\top H a$. This directly implies that the adversary should choose $a = H^{-1} \theta$ *regardless* of the sampled $x$.
* The above points imply that, once the learner selects the subset $T$ of features to ignore, there is no actual interplay between learner and adversary; the learner will set $\theta_T = 0$, choose $\theta_{-T}$ to minimize the MSE on the true distribution, and then incur loss $\mathbb{E} [(\theta^\top (x +H^{-1} \theta) - y)^2]$.
* It is clear, then, that the only interesting issue (at least from a computational complexity standpoint) would be how to choose the subset $T \subset [d]$ of features to ignore. However, the paper only considers the case where $d = 2$, so the learner has just three choices ($T = \lbrace \rbrace, \lbrace 1 \rbrace, \lbrace 2 \rbrace$).

Thus, the few results that the paper has follow immediately from straightforward calculations and are not particularly interesting.

**Suggestions for Authors:**

There are a lot of ways in which the paper could be made more theoretically appealing. The first ones that come to mind is to consider the high-dimensional setting (instead of $d=2$) and to equip the adversary with a more interesting or complex utility function, particularly one that makes the optimal manipulation $a$ be dependent on the feature vector $x$. Also, I do not feel that the additional figures presented in appendix B add much to the paper; the ones presented in the main body sufficiently illustrate the (somewhat obvious) point that it is safe not to drop features when manipulation is costly enough.

---

### Official Review · Reviewer_oqtH · 2025-09-18
**Nice results for 2-feature case, interested in seeing work for when multiple features should be dropped.**

**Rating:** 7
**Confidence:** 4

**Review:**

**Summary:**
- This paper provides a model to understand under what conditions a policymaker should drop features from use in a prediction model, when the features are potentially manipulable by agents. They are motivated by issues present in the U.S. Medicare Advantage, wherein private insurers are incentivized to misreport healthcare costs to receive a larger government payout. For the 2-feature case, they provide sufficient conditions for understanding when to drop a feature to preserve prediction accuracy.

**Strengths:**
- The paper is well motivated by a real-world example where features might be dropped to avoid manipulation.
- The optimization objectives are clearly defined, and the paper is overall well written.
- The graphs are helpful clearly illuminate the decision boundary.

**Weaknesses:**
- A primary weakness of this paper is that it only considers the 2-feature case, where the provider has the option to drop one of the features. - The author’s note on lines 134-135 that “feature correlation should guide feature selection decisions.” Therefore, it feels like studying the scenario where policymakers can strategically drop a subset of features would be very interesting.
- The results are also somewhat expected in the sense that my takeaway is that if manipulations are expensive, then you should keep both features.

**Suggestions:**
- I think it would be nice to have an additional example besides the CMS example to motivate when features should be dropped.

---

### Official Review · Reviewer_JcwC · 2025-09-20
**Offical review**

**Rating:** 5
**Confidence:** 3

**Review:**

* Summary

This paper analyzes feature selection in predictive models when input features can be strategically manipulated. The authors develop a linear regression framework with quadratic costs, derive sufficient conditions for when dropping manipulable features improves prediction risk, and identify threshold regimes that determine whether to retain or exclude features.

* Strengths

The work is clearly written, the proofs appear correct, and addresses an important problem in reliable ML under unreliable data. It contributes novel insights by linking feature selection decisions to strategic behavior, with results that are interpretable for real policy contexts (e.g., when audits or penalties make manipulation costly).

* Weaknesses / Limitations

The analysis relies on simplifying assumptions (linear regression), and results are mainly illustrated for two features. Simulations are provided but not supported by empirical case studies.
* Suggestions

The paper would be stronger with: (1) extensions to higher dimensions, (2) empirical validation with real or semi-synthetic data.